# A Secondary Analysis of the Complex Interplay between Psychopathology, Cognitive Functions, Brain Derived Neurotrophic Factor Levels, and Suicide in Psychotic Disorders: Data from a 2-Year Longitudinal Study

**DOI:** 10.3390/ijms25147922

**Published:** 2024-07-19

**Authors:** Pasquale Paribello, Mirko Manchia, Ulker Isayeva, Marco Upali, Davide Orrù, Federica Pinna, Roberto Collu, Diego Primavera, Luca Deriu, Edoardo Caboni, Maria Novella Iaselli, Davide Sundas, Massimo Tusconi, Maria Scherma, Claudia Pisanu, Anna Meloni, Clement C. Zai, Donatella Congiu, Alessio Squassina, Walter Fratta, Paola Fadda, Bernardo Carpiniello

**Affiliations:** 1Unit of Psychiatry, Department of Medical Sciences and Public Health, University of Cagliari, 09124 Cagliari, Italy; pasquale.paribello@unica.it (P.P.); diego.primavera@unica.it (D.P.); lucaderiu85@gmail.com (L.D.); massimotusconi@yahoo.com (M.T.);; 2Unit of Clinical Psychiatry, University Hospital Agency of Cagliari, 09124 Cagliari, Italy; 3Department of Pharmacology, Dalhousie University, Halifax, NS B3H 4R2, Canada; 4Division of Neuroscience and Clinical Pharmacology, Department of Biomedical Sciences, University of Cagliari, 09042 Cagliari, Italy; rcollu@bu.edu (R.C.); claudia.pisanu@unica.it (C.P.); squassina@unica.it (A.S.); wfratta@unica.it (W.F.); pfadda@unica.it (P.F.); 5Tanenbaum Centre for Pharmacogenetics, Campbell Family Mental Health Research Institute, Centre for Addiction and Mental Health, Toronto, ON M5T 1R8, Canada; clement.zai@camh.ca; 6Laboratory Medicine and Pathobiology, Department of Psychiatry, Institute of Medical Science, University of Toronto, Toronto, ON M5S 1A8, Canada; 7Centre of Excellence “Neurobiology of Dependence”, University of Cagliari, 09124 Cagliari, Italy

**Keywords:** psychotic disorders, suicidal behaviour, cognitive functions, BDNF

## Abstract

Identifying phenotypes at high risk of suicidal behaviour is a relevant objective of clinical and translational research and can facilitate the identification of possible candidate biomarkers. We probed the potential association and eventual stability of neuropsychological profiles and serum BDNF concentrations with lifetime suicide ideation and attempts (LSI and LSA, respectively) in individuals with schizophrenia (SCZ) and schizoaffective disorder (SCA) in a 2-year follow-up study. A secondary analysis was conducted on a convenience sample of previously recruited subjects from a single outpatient clinic. Retrospectively assessed LSI and LSA were recorded by analysing the available longitudinal clinical health records. LSI + LSA subjects consistently exhibited lower PANSS-defined negative symptoms and better performance in the BACS-letter fluency subtask. There was no significant association between BDNF levels and either LSI or LSA. We found a relatively stable pattern of lower negative symptoms over two years among patients with LSI and LSA. No significant difference in serum BDNF concentrations was detected. The translational viability of using neuropsychological profiles as a possible avenue for the identification of populations at risk for suicide behaviours rather than the categorical diagnosis represents a promising option but requires further confirmation.

## 1. Introduction

Suicide represents a complex health-related outcome deriving from an inextricable interaction of multiple factors comprising biological, neuropsychological, and environmental elements [1]. Each of these elements dynamically influences the impact of the remaining factors in determining suicide risk at a given time. Discrete differences in suicide rates exist across nations, with wide fluctuations over time, even in the same geographical location. Overall, while the absolute number of suicides has been increasing, the suicide rates have been decreasing with the concomitant increase in world population. The accuracy of suicide estimates also varies, depending on the cultural approach to death by suicide and on the existing death investigation infrastructure [1].

Psychiatric disorders represent a significant risk factor for suicide [2,3], with individuals living with schizophrenia (SCZ) featuring a particularly high risk of death by suicide and lifetime suicide attempts (4–10% and 25–50%, respectively) [4]. A public health approach to suicide would consider the transition from mental health disorders and suicide ideation to suicide attempts as linked phenomena lying on a continuum of human experience and behaviour [1]. Rates of transition from ideation to attempts vary widely, with reported figures suggesting that globally, they might range from 2.6 to 37% [5]. This high variability might depend on the relatively elevated clinical and biological heterogeneity of suicide. In addition, multiple cases of death by suicide may never receive a diagnosis or, indeed, never be in prior contact with mental health services [6,7,8].

Neurotrophins, especially brain-derived neurotrophic factor (BDNF), have been studied as possible candidate biomarkers for suicidal behaviours, with a recent review reporting promising results in post-mortem studies of individuals dead by suicide [9]. Through the interaction with the tyrosine receptor kinase B (TrkB), BDNF is believed to support neuron survival and differentiation and to modulate neurotransmission and synaptic plasticity in both central and peripheral nervous systems. Abnormalities in BDNF epigenetic regulation, transport, or signalling pathways have been linked to various neurological and psychiatric disorders [10,11]. Additionally, substantial evidence indicates that BDNF plays a crucial role in visceral pain and hypersensitivity conditions. The biological functions of BDNF are various and summarised in Figure 1 [12].

An additional systematic review summarised that BDNF concentrations appeared to be significantly associated with either recent or remote suicide attempts in plasma samples but not in serum samples [16]. Again, most studies of BDNF in suicide are cross-sectional, and only a few have included patients with SCZ [16]. In this context, we propose a secondary analysis of the Longitudinal Assessment of BDNF in Sardinian psychotic patient (LABSP) cohort [17], where we probed the possible stability of distinct psychopathological and cognitive profiles with specific patterns of serum BDNF concentration fluctuations among individuals with lifetime suicide ideation + lifetime suicide attempts vs. non-lifetime suicide ideation + lifetime suicide attempts and among individuals with lifetime suicide attempts vs. those without (LSI + LSA vs. non-LSI + LSA and LSA vs. non-LSA). Our null hypothesis was that there would be no difference between cognitive functions or the Positive and Negative Syndrome Scale (PANSS)-defined psychopathology for LSI and LSA and the rest of the sample.

## 2. Results

### 2.1. Sample Description

The recruitment process started in September 2014 and ended in March 2015. Consenting individuals were followed for two years starting from the date of signing the informed consent form. Specific to this project, lifetime suicide attempts and ideation (LSA and LSI, respectively) were coded only when there was evidence of some intent to die. Considering the retrospective nature of the assessment, the accuracy of the available information was not considered optimal for every involved subject in the original project. Data concerning LSI and LSA were considered adequate only for 88 subjects out of the original sample of 105 individuals. No suicide attempt was recorded during the 2-year follow-up period mandated by the study protocol. In the overall sample of 88 subjects considered for this report, 37 had a diagnosis of schizoaffective disorder (42.0%—SCA), whilst 51 had a diagnosis of schizophrenia (57.9%—SCZ). Nearly half of the overall sample had LSI (40 out of 88), with a subsample of LSI subjects also having LSA (25 out of the total 40 subjects featuring LSI). Considering how LSA was defined, all LSA subjects were coded as having also LSI. A summary of the sample’s major sociodemographic and clinical features at the baseline (T0) divided based on the LSI and LSA history is presented in Table 1. Figure 2 summarises in box-whiskers plots the fluctuations in letter fluency subtask performance, the PANSS-defined negative symptoms severity, and the LogBDNF serum concentrations between LSA + LSI vs. non-LSA + LSI, respectively. As expected, considering the naturalistic setting of this study, the prescribed treatments were numerous and comprised complex polytherapy regimens. In the overall sample of 88 patients included in the analysis, 16 (18.1%) received a depot medication, whilst 28 (31%) of subjects received clozapine at T0. Prescribed medications included haloperidol (19.3%), amisulpride (5.1%), aripiprazole (14.2%), chlorpromazine (3.8%), clozapine (22.0%), olanzapine (22.0), paliperidone (1.2%), quetiapine (6.4%), and risperidone (5.1%), with 21 subjects at T0 receiving more than one antipsychotic.

### 2.2. Association of LSI + LSA with Specific Psychometric and Neuropsychological Profiles and Peripheral BDNF

#### 2.2.1. LSI + LSA and PANSS-Defined Severity

A linear mixed model was used to probe the association between traditional PANSS subscales and PANSS subscales according to the pentagonal model with LSI + LSA, correcting for the effect of age at T0, time of the assessment, clozapine prescription, and education duration (years) (Table 2). We further examined the subgroup of LSA to explore the presence of a possible dose-effect in the associations eventually found, hypothesising that individuals with LSA might represent individuals featuring a higher severity within an ideation-to-action framework. Single PANSS item scores were also assessed using the same strategy. The results were the following:Negative Symptoms: Individuals with a history of LSI + LSA showed a significant reduction in PANSS-negative subscale severity, averaging 2.2 points lower than those without LSI + LSA (standard error = 0.623, z-score = −3.664, corrected *p*-value = 0.00124). This difference was even greater for the LSA-only subgroup, averaging 2.6 points lower (standard error = 0.875, z-score = −3.029, corrected *p*-value = 0.0123). Among LSA-only subjects, the average age was 0.11 years older than non-LSA subjects (standard error = 0.042, z-score = 2.719, corrected *p*-value = 0.0262);Judgment and Insight (G12): The severity of G12 (lack of judgment and insight) was significantly higher in LSA subjects, averaging 1.8 points higher (indicating worse judgment and insight) compared to those without LSA (standard error = 0.508, z-score = 3.580, corrected *p*-value = 0.001). This association was not significant for LSI + LSA (estimate = −0.255, standard error = 0.132, z-score = −1.929, corrected *p*-value = 0.269);Activation Subscale: The severity of the Activation subscale was significantly lower in LSI + LSA subjects, averaging 1.1 points lower (standard error = 0.420, z-value = −2.665, corrected *p*-value = 0.03). This association was only borderline significant for the LSA subgroup (estimate = −1.300, standard error = 0.538, z-score = −2.414, corrected *p*-value = 0.0643);Autistic Preoccupation Subscale: LSI + LSA subjects showed significantly lower scores on the Autistic Preoccupation subscale, averaging 1.4 points lower (standard error = 0.471, z-score = −3.125, corrected *p*-value = 0.00888). This was borderline significant for the LSA subgroup (estimate = −1.603, standard error = 0.638, z-score = −2.513, corrected *p*-value = 0.0599);Positive Symptoms scale—traditional subscale: The traditional positive subscale total score was significantly lower in LSA subjects, averaging 1.597 points lower (standard error = 0.625, z-score = −2.556, corrected *p*-value = 0.0424). These subjects were also younger (estimate = −0.086, standard error = 0.030, z-score = −2.829, corrected *p*-value = 0.0233) and had shorter education durations (estimate = −0.220, standard error = 0.088, z-score = −2.505, corrected *p*-value = 0.0424). No significant association was found for LSI + LSA subjects.

No significant associations were found for the pentagonal PANSS subscales of dysphoric mood and positive symptoms for either LSI + LSA or LSA patients.

#### 2.2.2. LSI + LSA and BACS-Defined Cognitive Performances

A linear mixed model was also applied to study BACS-defined cognitive performances raw data with LSA and LSI + LSA, correcting for the effect of education, time of the assessment, age, and clozapine therapy at T0 (Table 3). The results for the token subtask defined according to the BACS were unavailable due to limited data in the studied sample and were excluded from the present analysis. LSI + LSA subjects, on average, had 1.8 points higher in the BACS-letter fluency subtask than non-LSI + LSA (std. error = 0.50829, z-score = 3.580, corrected *p*-value Holm’s method = 0.00172). This association was, however, no longer statistically significant when considering only the LSA subjects (estimate = 0.71256, std. error = 0.55724, z-score = 1.279, corrected *p*-value Holm’s method = 0.726). Similarly, a trend of significance emerged between BACS-verbal memory subtask scores, with LSI + LSA scoring on average 0.5 higher points than non-LSI + LSA (estimate = 0.5910042, std. error = 0.2439477, z-score = 2.423, corrected *p*-value Holm’s method = 0.06163). Still, this association was no longer significant in the LSA subsample (estimate = −0.1538012, std. error = 0.2592079, z-score = −0.593, corrected *p*-value Holm’s method = 1.00000). No significant association emerged between LSA or LSI + LSA for the remaining BACS-defined subtasks.

#### 2.2.3. LSI + LSA and Serum BDNF Concentrations

A linear mixed model was applied to probe the possible association between the BDNF serum levels measured at different time points with LSI + LSA history, age, education, gender, and clozapine therapy (Table 4). The described model was significant for the effect of the timing of the assessment, but no significant association emerged for any of the included variables. Even when accounting for the presence of selected polymorphisms of the *BDNF* gene (rs11030104, rs1519480, rs7934165, rs6265—Val66Met), no significant association emerged between BDNF concentrations and LSI + LSA or LSA. The time point of the assessment was significantly associated with LogBDNF, suggesting a significant variation over time, with on average −0.01 lower LogBDNF level with each progressive time point (estimate = −0.012818, std. error = 0.002952, z-score = −4.342, corrected *p*-value Holm’s method = 8.48 × 10^−5^).

## 3. Discussion

In this report, we described the results of a secondary analysis for possible differences in the pattern of psychopathological and cognitive profiles and serum BDNF concentrations between LSI + LSA and non-LSI + LSA within a convenience sample of SCZ and SCA recruited in a single outpatient clinic over two years. LSI + LSA subjects showed lower levels of PANSS-defined negative symptoms and higher performances in BACS-defined verbal fluency tasks. We did not find any significant association for either LSA or for LSI + LSA with N1—blunted affect, G6—depressive symptoms, or any other PANSS subscale other than the classic negative PANSS subscale, activation, autistic preoccupation, insight and judgement subscale (G12), and a trend significance for the Pentagonal Negative Subscale. Our analysis adds to the existing literature by presenting the possible stability of the said neuropsychological profiles over 2 years [18]. The effect of insight on suicide risk in psychotic disorder appears unclear at this stage, with inconclusive evidence for an effect probably mediated by concomitant elements and possibly changing depending on the disease duration (e.g., first episode vs. successive ones) [19,20]. Cognitive abilities have been studied as components of insight. Research indicates that insight is more closely linked to memory and executive functions than overall intelligence. However, this association appears modest, implying that while intact neurocognitive abilities are necessary for insight, they do not constitute its core [20]. Here, we found relative preservation for some verbal fluency subtask performances in LSI + LSA but a worse insight and reality judgement in only LSA and not among LSI + LSA. Interestingly, albeit non-significant, when considering LSI + LSA, the insight and judgement subscale (G12) estimate was negative, suggesting a non-significant trend for better insight and judgement in this subsample. Considering the relative inconsistency of results when analysing LSI + LSA vs. LSA on its own for G12, it may be difficult to interpret these results. Indeed, the effect of insight may vary depending on the outcome considered. Similarly, we found a significant association between the BACS-letter fluency and verbal memory subtasks, but only for LSI + LSA, as this association was no longer significant when considering the LSA subsample. In the BACS-verbal memory subtask analysis, the non-significant estimate was of opposing signs among LSAs compared with LSI + LSA, substantiating the perplexities on the robustness of this finding for our sample. The association of cognitive performance patterns with suicidal ideation or attempts in the literature is contradictory, with several reports in SCZ finding evidence for a significant association for higher cognitive performances among attempters vs. non-attempters and others failing to replicate such findings [18]. A 2022 qualitative review on this topic [21] concluded that whilst the vast majority of studies in the field did not find an association between cognitive performances and LSA and LSI, three reported on the association of better performances among LSA and LSI, and three additional papers found evidence to the contrary. The evidence in SCA and other psychotic disorders is extremely scant in this regard, and no conclusion can be drawn in this population. The evidence for the association of positive or negative symptoms with LSI and LSA in SCZ is also conflicting, and no clear pattern has emerged [22]. In a previously published secondary analysis from the DNA Polymorphisms in Mental Illness Study (DPIM) involving 1494 participants, the study authors described a significant negative association between negative symptoms and a positive association of positive symptoms as defined according to the PANSS [23]. However, the association with positive symptom severity was no longer significant when accounting for the severity of depressive symptoms [23]. In our sample, we found that LSA had significantly lower positive symptom severity on average but not for LSI + LSA. Contrary to previous reports suggesting an association of PANSS-defined blunted effect with LSA, we did not find an association between this item and LSI + LSA or LSA in our sample [24]. Despite some significant associations in our study between specific psychopathological and cognitive profiles and LSI and LSA in SCZ, further research is needed to integrate these elements into clinical risk assessments for suicide. The current evidence is inconsistent, and the baseline risk for suicidal behaviours in high-risk populations remains low. Thus, even statistically significant associations might not translate into clinically meaningful risk increases. Predicting rare events like suicide might be particularly challenging, especially in the short term, making current suicide risk categorisation still far from clinical utility [25,26,27,28]. This should not lead to therapeutic or research nihilism but rather to focusing our collective efforts on researching and implementing preventive efforts with a robust evidence base. In fact, in non-emergency settings, stratifying populations at risk to target therapeutic efforts more accurately might represent a valuable option, considering the limited nature of resources in healthcare. Preliminary results in this setting suggest the possible worth of predictive instruments over long periods (i.e., years) [29,30]. The actionability of predictive models in emergency settings is complex due to limited evidence on the effectiveness of common interventions like acute hospitalisation in preventing suicide. While long-term risk assessment models might have potential, immediate predictions and interventions remain unreliable. This suggests a need to focus on preventive efforts with robust evidence bases, targeting high-risk individuals over extended periods. For instance, long-term therapeutic interventions analogous to those used in cardiology for myocardial infarction prevention could be applied to individuals at risk for suicide [25]. Empirical research is necessary to validate predictive instruments before they can be widely implemented. The heterogeneity of suicide phenotypes may have hindered progress in identifying candidate biomarkers and determining effective interventions. Identifying clinical phenotypes with specific risk determinants is crucial for advancing this line of research [31]. In this context, neurocognitive profile assessments could potentially play a role, though this remains to be clearly defined. Our study results also suggest fluctuations in peripheral BDNF levels over time, in line with past reports questioning the viability of BDNF as a stable clinical biomarker [12]. If the presented results regarding psychopathological and cognitive profiles in LSI + LSA subjects were to be replicated, the implications could be promising for the field. The definition of a clinical phenotype in psychotic disorders featuring a higher risk for suicide ideation or attempts could be employed for translational research in the pursuit of a clinically viable biomarker for this subpopulation of patients or to identify a specific population for testing psychotherapeutic interventions specifically aiming at suicide ideation.

### Limitations

Considering the retrospective nature of the assessment of LSI and LSA for our report, elements relevant to the assessment of suicidal behaviour and ideation might have been disregarded or incorrectly reported. The amount of time elapsed since the described events (either LSI or LSA) is often particularly significant for the recruited subjects. This latter element was not formally recorded and, therefore, was not available to be considered in this current study. Realistically, no inference can be drawn regarding the extension of the observed psychopathological profiles on individuals completing suicide, that is, among individuals ultimately dying by suicide [32]. This is particularly relevant to consider, especially because studies from violent death registries indicate that a sizeable portion of individuals ultimately dying by suicide do so at their first attempt, and often, these subjects have no preexisting, formally diagnosed mental disorders [33]. This population, despite representing a sizeable portion of suicide deaths at large, might be hard to study in clinical settings where retrospective studies investigating LSI and LSA are typically conducted. Moreover, even in clinical settings, past reports in the literature suggest that suicide completers may have a different profile altogether from suicide attempters [32], despite the latter representing a standalone risk factor for death by suicide [34]. In the present report, we did not find a significant association for serum BDNF level concentrations over a 2-year follow-up period with LSA or LSI. Our sample size might have been insufficiently powered to detect the presence of moderate BDNF differences in this regard. This represents a secondary analysis for a previously recruited sample, which might not have enough power to detect an effect even if present [17]. Moreover, BDNF peripheral level variations associated with LSI and LSA might have occurred more proximal to the described events and may, therefore, be undetectable at later times. We did not collect plasma samples to assess BDNF concentration, which may also represent an additional limitation to our report. In fact, a previous systematic review concluded that BDNF in plasma but not in serum appeared significantly lower among individuals with LSA [16]. However, these results need to be interpreted with due consideration of the relative instability of plasma BDNF assessment and their relatively low retest stability over a year, with certain authors suggesting the use of serum samples over plasma for this reason [12]. No pro-BDNF assessment was performed. Therefore, we cannot assess its worth as a biomarker candidate in our sample [35]. It is also impossible to correct our findings for eventual changes in the platelet BDNF levels, as no platelet BDNF level was assessed. No data were collected regarding the menstrual cycle phase, and we cannot correct its effect on female subjects [36]. However, the results appear unchanged even when only data from male subjects from our sample were analysed. The central BDNF correlation with peripheral BDNF concentrations remains unclear, with evidence both in favour and against the potential viability of peripheral BDNF as a candidate biomarker for mental disorders [12]. In sum, we did not find a persistent pattern of serum BDNF concentrations that could distinguish subjects with LSI from those with LSA history. However, we did find evidence of the persistence of over 2 years of differing psychopathological profiles in this sample, with LSI + LSA associated with a lower negative PANSS subscale severity and higher performances in the BACS-letter fluency subtask.

## 4. Materials and Methods

### 4.1. Design and Setting

This is a secondary analysis of a Primavera 2017 [17,37]. A sample of 105 patients was recruited from the Clinic of Psychiatry in Cagliari, selected based on the presence of a DSM-IV-TR diagnosis of either schizophrenia or schizoaffective disorder formulated by the responsible psychiatrist and confirmed by responsible research personnel at the Clinic of Psychiatry in Cagliari. The Clinic of Psychiatry in Cagliari is an outpatient psychiatric clinic within the national healthcare system located in the metropolitan area of Cagliari, in the south of Sardinia. It offers community mental healthcare to an area inhabited by approximately 70,000 individuals, with nearly 2400 subjects regularly receiving care regularly. Inclusion criteria for the original study were as follows: (1) age ranging from 18 to 65 years old; (2) diagnosed with schizophrenia (SCZ) or schizoaffective disorder (SAD) according to Diagnostic and Statistical Manual of Mental Disorders, 4th ed. (DSM-IV-TR); and (3) stable condition for the past six months before recruitment. The exclusion criteria were (1) refusal to provide consent, (2) presence of acute psychopathological symptoms, (3) cognitive impairment due to illness severe enough to hinder cooperation, (4) major unstable medical illness, (5) severe mental retardation, (6) major neurological disorder or previous traumatic brain injury, and (7) current drug or alcohol dependence. No data were available regarding the number of individuals eventually contacted but excluded for failing to meet inclusion criteria or refusing to provide consent.

### 4.2. Clinical Assessments

The recruited subjects were required to donate a blood sample at baseline and at four additional time points scheduled every six months for the following two years. Contextually, they were formally assessed with the use of (1) the Positive and Negative Symptoms Scale (PANSS), (2) Clinical Global Impression Schizophrenia (CGI-SCH), (3) Brief Assessment of Cognition in Schizophrenia (BACS), (4) the Personal and Social Performance Scale (PSP), (5) Subjective Wellbeing under Neuroleptics-Short Version (SWN-S), and (6) the WHO Quality of Life brief questionnaire (WHOQOL-Brief). No reimbursement or financial incentive was available for the recruited individuals during the project. Data on personal and family psychiatric history and a detailed treatment history were also recorded. The presence of LSA was assessed through the analysis of the available clinical health records at our institution and, when available, through clinical interviews. Suicide attempts were defined based on the presence of at least a potentially lethal attempt with a clear intention to die. Considering the retrospective nature of this analysis, only the subjects for which this element was available were included in the report.

### 4.3. Statistical Analysis

Statistical analyses were performed using JASP 0.16.3.0 and R studio Team, 2020. For this project, we employed (1) the Wilcoxon test for parametric variables and (2) linear mixed-effects models to assess the eventual associations between LSI + LSA and the studied clinical elements during the 2-year follow-up period. The model was corrected for the effect of age, education duration and the presence of clozapine therapy. Clozapine, among the others, was felt to be particularly relevant to our analysis in consideration of its unique efficacy profile and the relative indication for this medication in individuals with suicide behaviours [38]. Considering the relatively limited dimension of the studied sample, no specific analysis could be conducted either for other specific medications or specific combinations of medications. Log transformation was used for peripheral BDNF concentrations. Holm’s method was employed to correct the *p*-value for the effect of multiple comparisons on the significant associations eventually found.

### 4.4. Sampling and BDNF Assessment

Blood samples were collected from each patient between 8:00 and 10:00 a.m. The BDNF ELISA Kit was employed to assess the serum BDNF levels (Booster Immunoleader Biological Technology Co., Ltd., Pleasanton, CA, USA, catalogue no EK0307), specifically designed for the quantitative detection of human BDNF in cell culture supernatants, serum, and plasma. This kit employs standard sandwich ELISA technology to ensure precise quantification of natural and recombinant human BDNF, boasting high sensitivity (<2 pg/mL) and no detectable cross-reactivity with other relevant proteins. Upon blood collection, serum samples were allowed to clot in a serum separator tube at room temperature for approximately 4 h before being centrifuged at around 1000× *g* for 15 min. Following centrifugation, supernatant serum samples were divided into small aliquots and promptly stored at −20 °C for future analysis. Subsequently, samples underwent processing in accordance with the kit protocol and instructions provided. To measure the optical density absorbance of each sample, a microplate reader (Thermo Scientific Multiskan FC, Thermo Fisher Scientific Oy Ratastie 2, Vantaa, Finland) equipped with a 450 nm filter was employed within 30 min following the completion of the kit procedure. The data obtained were analysed using Thermo Scientific SkanIt Software V.3.0 for Multiskan FC.

## 5. Conclusions

In a sample of individuals with SCZ and SCA, we described the relative persistence over a 2-year follow-up period of differing verbal fluency subtask performances and PANSS-defined negative symptoms between LSI + LSA vs. non-LSI + LSA. No specific serum BDNF concentration fluctuation pattern was observed to discriminate between these two subpopulations. Additional interesting findings with either traditional or pentagonal-defined PANSS subscales, while less robust, warrant further investigations.

## Figures and Tables

**Figure 1 ijms-25-07922-f001:**
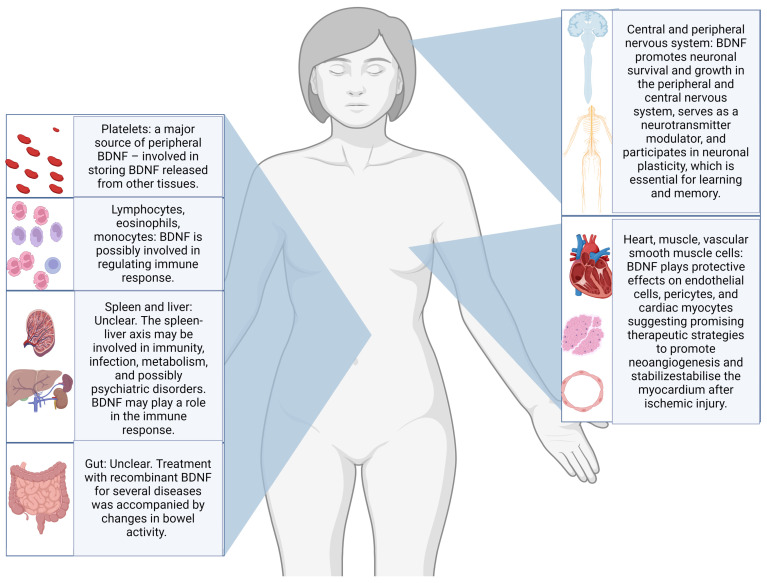
Illustration of the most frequently reported body sites for production and storing, as well as some of the postulated biological functions of BDNF [12,13,14,15].

**Figure 2 ijms-25-07922-f002:**
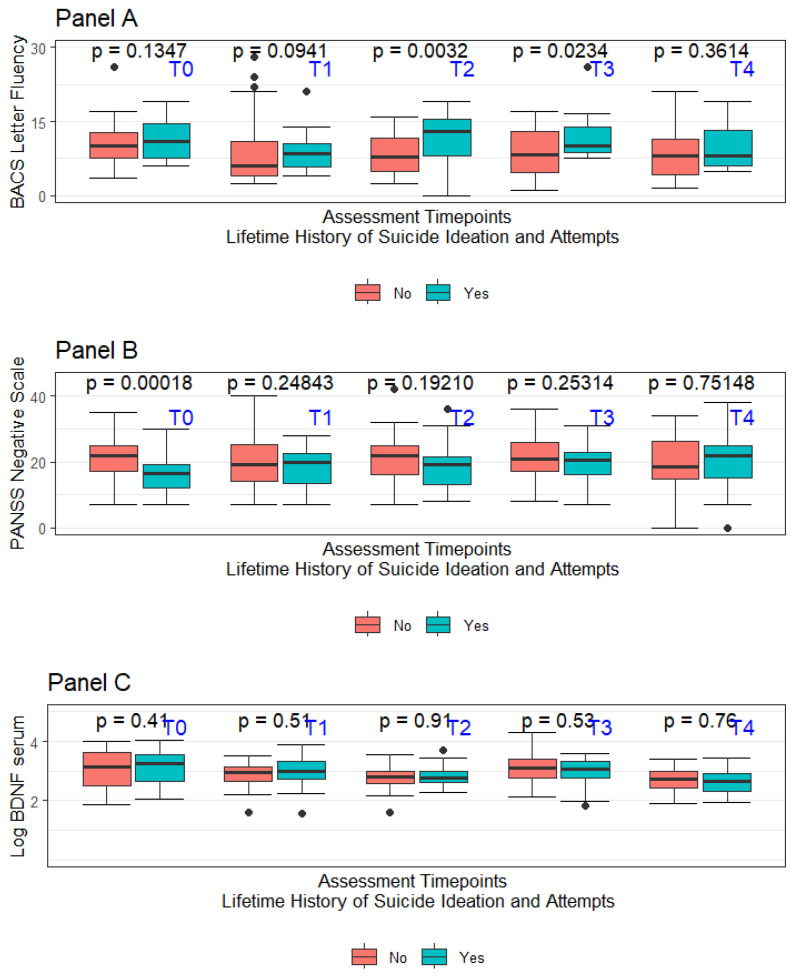
**Panel A**—Box-whiskers plot for BACS letter fluency divided per assessment time for LSI + LSA vs. non-LSI + LSA; **Panel B**—Box-whiskers plot for PANSS-Negative traditional subscale divided per assessment time for LSI + LSA vs. non-LSI + LSA; **Panel C**—Box-whiskers plot for BDNF serum levels divided per assessment time for LSI + LSA vs. non-LSI + LSA.

**Table 1 ijms-25-07922-t001:** Principal sociodemographic and clinical features for the included sample at T0.

Studied Variable (T0)	LSI (n = 40)	LSA (Subsample of LSI; n = 25)	Non-LSI or LSA (n = 48)	*p*-Value (Comparisons Are for LSI vs. Non-LSI or LSA.) ^1^
Age–years, median (25th–75th percentile)	46.5 (39.0–52.2)	46.0 (39.0–51.0)	47.5 (42.0–57.0)	0.0251
Female sex (n—%)	8 (9.0%)	4 (4.5%)	18 (20.4%)	0.073
Education–years, median (25th–75th percentile)	8.0 (8.0–13.0)	8.0 (8.0–13.0)	8.0 (8.0–13.0)	0.803
Civil status (n—%)				0.293
Single	5 (5.6%)	0 (0.0%)	3 (3.4%)	
Married/Cohabiting	2 (2.2%)	2 (2.2%)	6 (6.8%)	
Divorced	0 (0.0%)	0 (0.0%)	1 (1.1%)	
Widowed	33 (37.5%)	23 (26.1%)	36 (40.9%)	
Not available	0 (0.0%)	0 (0.0%)	2 (2.3%)	
Employment status (n—%)				0.051
Employed	6 (6.8%)	4 (18.1%)	1 (1.1%)	
Housewife	0 (0.0%)	0 (0.0%)	0 (0.0%)	
Student	0 (0.0%)	0 (0.0%)	1 (1.1%)	
Retired	0 (0.0%)	0 (0.0%)	0 (0.0%)	
Registered disabled civilians	33 (37.5%)	21 (23.8%)	45 (51.1%)	
Unemployed	0 (0.0%)	0 (0.0%)	2 (2.2%)	
Diagnosis SCZ (n—%)	24 (27.2%)	14 (15.9%)	27 (30.6%)	0.723
Diagnosis SCA (n—%)	16 (18.1%)	11 (12.5%)	21 (23.8%)	0.874
Past hospital admissions (n—%)	34 (38.6%)	22 (25.0%)	44 (50.0%)	0.326
Average age of onset, years–median (25th–75th percentile)	19.5 (18.0–23.2)	20.0 (18.0–23.0)	20.0 (16.0–28.0)	0.743
Average illness duration, months–median (25th–75th percentile)	306.0 (192.0–384.0)	288.0 (192.0–384.0)	312.0 (189.0–375.0)	0.594
Duration of untreated psychosis, months–median (25th–75th percentile)	7.0 (1.0–24.0)	24.0 (2.0–24.0)	6.0 (1.0–36.0)	0.852
Body Mass Index–median (25th–75th percentile)	25.7 (21.5–31.2)	25.4 (21.7–30.8)	26.9 (23.5–31.0)	0.397
Long-acting injectable antipsychotic therapy (n—%)	5 (5.6)	5 (5.6)	11 (12.5)	0.175
Clozapine therapy (n—%)	13 (14.7)	7 (7.9)	15 (17.0)	0.900
WHO Quality of Life-BREF–physical health, median (25th–75th percentile), IQR)	12.5 (10.4–14.2)	12.8 (10.7–14.2)	13.1 (10.8–14.8)	0.505
WHO Quality of Life-BREF–psychological health, median (25th–75th percentile)	11.3 (10.6–12.6)	11.3 (10.6–12.6)	12.0 (11.3–13.2)	0.117
WHO Quality of Life-BREF–social relationships, median (25th–75th percentile)	11.3 (9.3–13.3)	11.3 (9.3–13.6)	10.6 (8.3–13.3)	0.821
WHO Quality of Life-BREF–environmental health, median (25th–75th percentile)	11.5 (10.1–13.0)	11.5 (10.3–13.0)	12.5 (11.0–14.0)	0.089
Personal and Social Performance Scale (PSP)—total score, median (25th–75th percentile)	55.0 (45.0–65.0)	50.0 (45.0–65.0)	47.5 (40.0–60.0)	0.075
Subjective Wellbeing under Neuroleptics-Short Version (SWN-S)—total score, median (25th–75th percentile)	81.0 (70.0–93.0)	82.0 (73.5–90.0)	80.0 (67.2–85.0)	0.246
Clinical Global Index–Schizophrenia Overall Severity, median (25th–75th percentile)	3.0 (3.0–4.0)	3.0 (3.0–4.0)	4.0 (3.0–4.0)	0.017

^1^ We used T-Student for normally distributed variables, the Kruskal–Wallis for non-normally distributed ones, and χ^2^ for categorical variables. Comparisons are for LSI + LSA vs. non-LSI + LSA.

**Table 2 ijms-25-07922-t002:** Selected PANSS subscales scores for the included sample at T0.

Positive and Negative Symptoms Scale (PANSS—T0)	LSI (n = 40)	LSA (Subsample of LSI n = 25)	Non-LSI or LSA (n = 48)	*p*-Value (Comparisons are for LSI vs. Non-LSI or LSA) ^1^
Positive scale traditional–total score, median (25th–75th percentile)	13.0 (11.0–15.2)	13.0 (11.0–16.0)	16.0 (12.0–18.0)	0.039
Negative scale traditional–total score, median (25th–75th percentile)	16.5 (12.0–19.2)	18.0 (15.0–20.0)	22.0 (17.0–25.0)	0.001
General scale traditional–total score, median (25th–75th percentile)	37.5 (29.0–41.7)	38.0 (31.0–44.0)	45.0 (37.7–49.2)	0.005
Positive scale pentagonal model–total score, median (25th–75th percentile)	11.0 (9.0–12.2)	11.0 (10.0–12.0)	12.5 (10.0–14.2)	0.041
Negative scale pentagonal model–total score, median (25th–75th percentile)	20.5 (14.7–25.2)	23.0 (20.0–27.0)	28.0 (22.7–31.2)	0.001
Autistic preoccupation pentagonal model–total score, median (25th–75th percentile)	14.0 (11.0–16.2)	15.0 (13.0–17.0)	17.0 (14.0–20.2)	0.001
Dysphoric mood pentagonal model–total score, median (25th–75th percentile)	12.0 (9.7–15.2)	12.0 (10.0–16.0)	13.0 (10.0–16.0)	0.447
Activation pentagonal model–total score, median (25th–75th percentile)	14.0 (9.0–13.0)	11.0 (10.0–13.0)	13.0 (10.7–16.0)	0.002

^1^ We employed the T-Student for normally distributed variables and the Kruskal–Wallis for non-normally distributed ones. Comparisons are for LSI + LSA vs. non-LSI + LSA.

**Table 3 ijms-25-07922-t003:** BACS subtasks performances at T0.

Brief Assessment of Cognition in Schizophrenia (BACS—T0)	LSI (n = 40)	LSA (Subsample of LSI n = 25)	Non-LSI or LSA (n = 48)	*p*-Value (Comparisons Are for LSI vs. Non-LSI or LSA) ^1^
BACS–verbal memory–total score, median (25th–75th percentile)	5.8 (4.4–7.8)	6.8 (5.0–9.7)	5.9 (5.0–8.1)	0.444
BACS–letter fluency–total score, median (25th–75th percentile)	11.0 (7.5–14.5)	13.0 (10.0–15.5)	10.0 (7.5–12.7)	0.136
BACS–digit sequencing test—total correct, median (25th–75th percentile)	14.0 (11.0–16.0)	15.0 (13.5–17.2)	13.5 (11.0–16.7)	0.666
BACS–digit sequencing test—longest correct sequence, median (25th–75th percentile)	5.0 (4.0–7.0)	6.0 (5.0–7.0)	5.0 (4.0–7.0)	0.111
BACS–symbol coding–total score, median (25th–75th percentile)	26.0 (20.0–32.0)	30.5 (26.7–44.7)	29.5 (23.5–46.5)	0.030
BACS–Tower of London–total score, median (25th–75th percentile)	9.0 (5.0–14.0)	10.5 (5.7–17.0)	13.0 (9.0–16.0)	0.030

^1^ We employed the T-Student for normally distributed variables and the Kruskal–Wallis for non-normally distributed ones. Comparisons are for LSI + LSA vs. non-LSI + LSA.

**Table 4 ijms-25-07922-t004:** Serum BDNF concentrations at T0.

	LSI (n = 40)	LSA (Subsample of LSI n = 25)	Non-LSI or LSA (n = 48)	*p*-Value (Comparisons Are for LSI vs. Non-LSI or LSA) ^1^
Serum BDNF concentrations ng/mL, median (25th–75th percentile)	25.490 (14.200–35.455)	21.380 (13.170–33.700)	23.115 (12.160–37.160)	0.482

^1^ We employed the Kruskal–Wallis test. The comparison is for LSI + LSA vs. non-LSI + LSA.

## Data Availability

Data are contained within the article.

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
