# Peer review of "A Secondary Analysis of the Complex Interplay between Psychopathology, Cognitive Functions, Brain Derived Neurotrophic Factor Levels, and Suicide in Psychotic Disorders: Data from a 2-Year Longitudinal Study"

_ijms, 2024, doi:10.3390/ijms25147922_

Round 1

Reviewer 1 Report

Comments and Suggestions for Authors

To the AAs

In the search of identifying phenotypes at high-risk of suicidal behaviour and potential candidate biomarkers, the present Ms tested a potential association between serum BDNF concentrations and suicidal behavior (LSI and LSA) in SCZ and SCA patients (conditions known to be at high risk of LSI/LSA) from a 2-yr-long follow-up study.  The results of this retrospective study did not reveal a significant association between serum BDNF concentrations and suicidal behavior in the examined study populations.  The relevance of the main findings and AAs’ conclusions are unclear and quite obscure to this reviewer (“However, we found evidence for the relative persistence of lower PANSS-defined negative symptoms and higher performances in the BACS-literal fluency subtask in LSI+LSA subjects. Conclusions: We found a relatively stable pattern of lower negative symptoms over 2 years among patients with LSI and LSA”; Abstract, lines 35-38).

Although the rationale of the study has some interest, I personally can see several major shortcomings regarding text prolixity, study design, use of statistical tools, results description, data interpretation and relevance of conclusions.

Criticism Points

1.      Introduction: the section preceeding the study rationale (i.e., apx. 40 lines) could be safely shortened. I would encourage the Authors (AAs) to make an effort with aim of improving both Ms readability and interest to the readership;

2.      Lines 93-94 “Abnormalities in BDNF’s epigenetic regulation, transport, or signalling pathways have been linked to various neurological and psychiatric disorders”. Please include the corresponding references;

3.      Table 1. Instead of a single figure IQR should be better expressed as 25th to 75th quartiles intervals, as usual;

4.      It is difficult to understand the use of Student t-tests in a 3-groups comparison. Please clarify;

5.      The PANSS acronym is not explained in the text. Please clarify

6.      The Results section is quite lengthy and confusing. I would encourage the AAs to made efforts towards making it more readable and clearer;

7.      The Discussion section is very verbose, confusing and quite difficult to read.  I would encourage the AAs to made efforts towards making it more readable;

8.      The language, although usually acceptable, needs to be re-edited for the sake of a better concept clarity and more succint presentation;

Comments on the Quality of English Language

The language, although usually acceptable, needs to be re-edited for the sake of a better concept clarity and more succint presentation. 

Author Response

R) We thank the reviewer for the positive assessment of our work.
Q1) Introduction: the section preceding the study rationale (i.e., apx. 40 lines) could be safely
shortened. I would encourage the Authors (AAs) to make an effort with aim of improving both Ms
readability and interest to the readership.
R1) Thank you for your feedback. The introduction section was substantially shortened and
revised as suggested by the reviewer.
Q2) Lines 93-94 “Abnormalities in BDNF’s epigenetic regulation, transport, or signalling pathways
have been linked to various neurological and psychiatric disorders”. Please include the
corresponding references.
R2) We added references supporting this statement.
Q3) Table 1. Instead of a single figure IQR should be better expressed as 25th to 75th quartiles
intervals, as usual.
R3) Tables 1,2 and 3 were edited to reflect the requested changes.
Q4) It is difficult to understand the use of Student t-tests in a 3-groups comparison. Please clarify.
R4) Thank you for your feedback. We copied the description for the p-value from the footnote to
the column label to improve its clarity for all the three tables of the manuscript (i.e., the p-value
refers to the comparison LSI+LSA vs non-LSI+LSA, first vs third column).
Q5) The PANSS acronym is not explained in the text. Please clarify.
R5) As requested, we spelled out the PANSS acronym in the manuscript.
Q6) The Results section is quite lengthy and confusing. I would encourage the AAs to made
efforts towards making it more readable and clearer.
R6) Thank you for your feedback. The results section regarding the PANSS sub analysis was
revised and reorganized in a bullet-point fashion to improve its clarity. We added additional
feedback on the rational for the analysis.
Q7) The Discussion section is very verbose, confusing and quite difficult to read.  I would
encourage the AAs to made efforts towards making it more readable.
R7) Thank you for your feedback. We revised extensively the discussion section, and as a result,
from 1274 words, it is now 1049 words, whilst keeping the same number of references and general
meaning. As requested by reviewer 2, we added additional feedback on the relevance of the
presented data and possible future direction whilst keeping a reduction in the overall words, as
requested.

Q8) The language, although usually acceptable, needs to be re-edited for the sake of a better
concept clarity and more succinct presentation.
R8) The manuscript was extensively revised to reflect the requested changes.

Reviewer 2 Report

Comments and Suggestions for Authors

Paribello et al. performed a secondary analysis on a convenience sample of previously recruited subjects from a single outpatient clinic. Retrospectively assessed LSI and LSA were recorded by analysing the available longitudinal clinical health records. This is an interesting paper, indicating that a relatively stable pattern of lower negative symptoms over 2 years among patients with LSI and LSA, but no significant difference in serum BDNF concentrations, possible candidate biomarkers for suicide behaviors, was detected. There are, however, several issues to be addressed to further improve the manuscript.

1.     Although the authors mentioned a "convenience sample of previously recruited subjects from a single outpatient clinic." , more detail about the sample size, demographic characteristics, and selection criteria would enhance clarity and context.

2.     Although this is a 2-yeear follow-up study, the time period from SCZ or SCA onset needs to be clarified.

3.     The conclusions section suggests that neuropsychological profiles may be a viable avenue for identifying at-risk populations. Discussing specific implications for clinical practice or future research directions would strengthen the paper.

Author Response

R) We thank the reviewer for the positive assessment of our work.
Q1) Although the authors mentioned a "convenience sample of previously recruited subjects from
a single outpatient clinic.", more detail about the sample size, demographic characteristics, and
selection criteria would enhance clarity and context.
R1) Thank you for your feedback. An additional description was added to the methods section to
expand the characterization of our sample further:” Inclusion criteria for the original study are as
follows: (1) age ranging from 18 to 65 years old; (2) diagnosed with schizophrenia (SCZ) or
schizoaffective disorder (SAD) according to Diagnostic and Statistical Manual of Mental
Disorders, 4th ed. (DSM-IV-TR); (3) stable condition for the past six months before recruitment.
Exclusion criteria are: (1) refusal to provide consent; (2) presence of acute psychopathological
symptoms; (3) cognitive impairment due to illness severe enough to hinder cooperation; (4) major
unstable medical illness; (5) severe mental retardation; (6) major neurological disorder or
previous traumatic brain injury; (7) current drug or alcohol dependence. No data is available
regarding the number of individuals eventually contacted but who were ex-cluded for failing to
meet inclusion criteria or refusing to provide consent.”
Q2)     Although this is a 2-year follow-up study, the time period from SCZ or SCA onset needs to be
clarified.
R2) Thank you for your feedback. Additional rows in Table One now describe the average age of
onset, the duration of untreated psychosis and the average duration of disease for all the three
groups (i.e., individuals with Lifetime suicide ideation-LSI, attempts-LSA and non-LSI or LSA).
Q3)   The conclusions section suggests that neuropsychological profiles may be a viable avenue
for identifying at-risk populations. Discussing specific implications for clinical practice or future
research directions would strengthen the paper.
R3) Thank you for your kind feedback. To address the need to discuss the implications of these
results further whilst keeping in line with the requirement to summarise our draft from reviewer 1,
we added the following passage to the discussion section, still resulting in a significant reduction
in the discussion section length (i.e., from 1274 words to 1049 words) “If the presented results
regarding psychopathological and cognitive profiles in LSI+LSA subjects were to be replicated,
the implications could be promising for the field. The definition of a clinical phenotype in

psychotic disorders featuring a higher risk for suicide ideation or attempts could be employed for
translational research in the pursuit of a clinically viable biomarker for this subpopulation of
patients or to identify a specific population for testing psychotherapeutic interventions
specifically aiming at suicide ideation.”

Round 2

Reviewer 1 Report

Comments and Suggestions for Authors

To the AAs

The article is substantially improved in terms of readability. Although some mispelling and/or style imperfections could still be spotted around, the language style is much improved. The AAs made major efforts in carefully addressing all my constructive criticism points. Although study limitations remain substantially unchanged, the Ms is, in my opinion, ready to open debate from the scientific community provided that a minor text editing would be guaranteed.

Comments on the Quality of English Language

Some mispellings and/or style imperfections still require a MINOR text editing

Author Response

The article is substantially improved in terms of readability. Although some mispelling and/or style imperfections could still be spotted around, the language style is much improved. The AAs made major efforts in carefully addressing all my constructive criticism points. Although study limitations remain substantially unchanged, the Ms is, in my opinion, ready to open debate from the scientific community provided that a minor text editing would be guaranteed.

R: Thank you for the overall positive judgement on the manuscript. We further edited it to address inaccuracies and misspellings.

Reviewer 2 Report

Comments and Suggestions for Authors

The authors put an effort in revising their manuscript and addressing issues raised previously. The paper was improved.

Author Response

The authors put an effort in revising their manuscript and addressing issues raised previously. The paper was improved.

R: Thank you for the favorable judgement on our manuscript. I believe that the reviewing process significantly improved our draft.